# CORRESPONDENCE

# The role of MHC supertypes in promoting trans-species polymorphism remains an open question

Maciej J. Ejsmond[1,2], Karl P. Phillips[3,4], Wiesław Babik[1] & Jacek Radwan [5]

In a recently published paper on the evolution of the vertebrate major histocompatibility complex (MHC), Lighten et al.[1]; (henceforth 'Lighten et al.') set out to explain an apparent incompatibility between the dynamic nature of host-parasite coevolution, which could accelerate MHC allele turnover[2,3], and the apparent long-term persistence of allelic lineages that may underpin MHC trans-species polymorphism (TSP[4,5]). TSP arises when multiple allelic lineages that originated in an ancestral species are maintained in descendant species. TSP is usually a transient phenomenon, but at the MHC, TSP is common and seemingly long-term, leading to profound discordance between genealogies of MHC alleles and species phylogenies[4,5]. Lighten et al. offer a scenario in which several functionally divergent MHC 'supertypes' (clusters of MHC alleles with similar physicochemical properties at their antigen-binding sites[6]) are maintained by balancing selection, whereas functionally similar alleles within supertypes undergo fast turnover. The scenario is based on an empirical finding that population-genetic structure by supertypes is significantly lower than allele-based null expectations, and on simulations modelling MHC alleles as coordinates in paratope space. Here, we argue that the empirical patterns do not support a major role of supertypes in the maintenance of TSP, and that the theoretical arguments provided by Lighten et al. are based on disputable assumptions.

TSP is a feature of gene genealogy and requires phylogenetic analysis to demonstrate it. In a phylogeny, TSP is detected as monophyletic groups of alleles with each group represented in multiple species, or by extensive paraphyly should some lineages be lost in one or more descendent species. Despite the complexities of molecular evolution (recombination, gene conversion) that often cause departures of the true genealogy from a bifurcating tree, such TSP-diagnostic patterns are commonly detected in the MHC[5,7]. If TSP is caused by long-term stability of MHC supertypes, the phylogeny of MHC alleles from two sister species should be characterized by predominantly monophyletic supertypes present in both species (Fig. 1a), barring occasional

supertype loss in either species. The tree Lighten et al. present in their Supplementary Fig. 2 is based on amino acid sequences at 15 codons under significant positive selection. Although such targeted trees are useful in studying MHC evolution, longer sequences will provide a more rigorous test of a phylogenetic property such as TSP. We therefore constructed a phylogeny based on Lighten et al.'s full nucleotide sequences (Fig. 1b). We did not observe the predicted pattern: most supertypes were far from monophyletic, which suggests convergent origin rather than common ancestry of alleles within supertypes. Lighten et al. argue that gene conversion between alleles of different supertypes has broken down monophyly for all supertypes except ST9 (legend to Supplementary Fig. 2 in ref. [1]), but do not explain how polyphyly of supertypes can be reconciled with their proposed role in TSP. Instead, we think that different allelic clades of the same supertype would readily fix in different species, even under strong selection maintaining supertypes themselves, thus erasing TSP.

Our attention was drawn by ST9, the only supertype that forms a well-supported clade in the phylogenetic tree (Fig. 1b). Divergence within this clade is relatively shallow, indicating rapid coalescence, and a long branch separates ST9 clade from other MHC alleles. This pattern is strikingly different from the rest of the tree (Fig. 1b). Lighten et al. note a correlation between the number of ST9 alleles and the number of microsatellite alleles within populations and suggest that ST9 alleles are subject to drift. Thus, ST9 may represent a specialized MHC locus that evolves differently from other MHC supertypes. In support of this, alleles comprising ST9 map best to a different scaffold in the guppy genome (LG18: 22 Mb) than alleles of the other supertypes: (unplaced scaffold 55 kb or unplaced scaffold 1.5 kb). Importantly, our analyses showed that the central observation of Lighten et al.—of weaker population-genetic structuring of supertypes compared to structuring of MHC alleles—appears to be entirely driven by ST9, as indicated by jack-knife removal of each supertype (Supplementary Figs 1, 2). The effect of ST9 does not appear to be solely due to its high frequency in the dataset

[1] Institute of Environmental Sciences, Jagiellonian University, ul. Gronostajowa 7, 30-387 Kraków, Poland. [2] Centre for Ecology and Evolution in Microbial model Systems – EEMiS, Linnaeus University, 39182 Kalmar, Sweden. [3] School of Biological, Earth & Environmental Sciences, University College Cork, Cork T23 N73K, Ireland. [4] Marine Institute, Furnace, Newport, Co. Mayo F28 PF65, Ireland. [5] Evolutionary Biology Group, Institute of Environmental Biology, Adam Mickiewicz University, ul. Umultowska 89, Poznan, Poland. Correspondence and requests for materials should be addressed to J.R. (email: jradwan@amu.edu.pl)

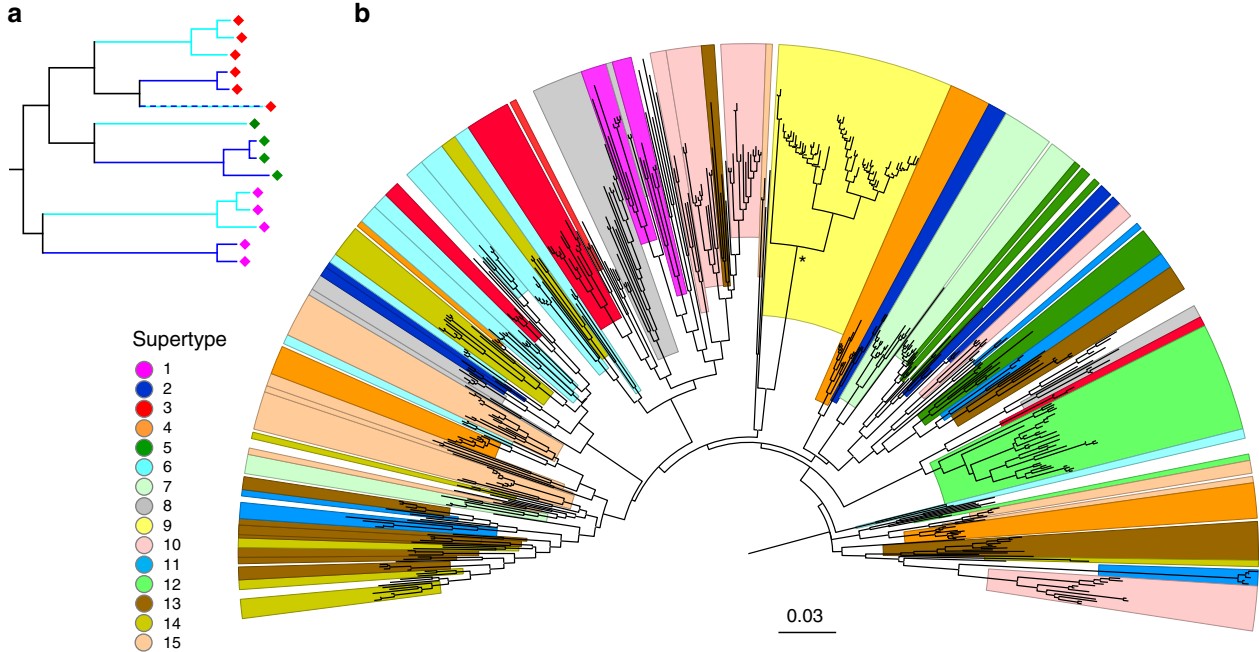

**Fig. 1** Predictions and results of phylogenetic analysis. **a** The hypothetical pattern expected if balancing selection on supertypes maintains TSP, but alleles within supertypes experience rapid turnover. Hypothetical supertypes are marked with diamonds of the same colour, and the species-specific branches are in turquoise and deep blue. Supertype lineages should be shared between species, but sharing of identical alleles (dashed line) should be rare. **b** Neighbour-joining tree of 539 MHC class II exon 2 alleles reported by Lighten et al.[1] The tree was constructed from a matrix of Jukes–Cantor distances calculated using all 209 nucleotide positions. Alleles from each supertype are marked with the same colour as in Lighten at al. Supertype 9 is marked yellow; the asterisk indicates that it is the only monophyletic supertype (bootstrap support 97%)

(Supplementary Table 1), which further suggests that this supertype may be experiencing different evolutionary pressures from the other supertypes. The lower genetic structuring of supertypes is therefore not a general feature of the dataset. Another argument which Lighten et al. used to support balancing selection acting on supertypes, a deficit of "supertype homozygotes", is unconvincing because the authors could not assign alleles to loci (genotypes consisted of up to nine alleles, implying a minimum of five co-amplifying loci in some individuals), meaning that true zygosity is not known. Furthermore, the unique properties of ST9 also mean that the observed homozygosity deficit needs to be treated cautiously (6.7% of individuals are ST homozygotes in the full dataset, but 29.1% if ST9 and all its alleles are removed).

The theoretical argument of Lighten et al. is based on simulations suggesting that supertypes may persist nearly indefinitely without much change in their antigen-binding properties, shown by stable positions of supertypes in paratope space. In contrast, alleles persist for very short times, being replaced by new mutant alleles within the same supertype (Fig. 4 of Lighten et al.). While the numbers of supertypes (~ 8) and alleles (~ 40 in a subsample of 100 individuals) in the simulated populations of Lighten et al. are comparable to those observed in natural populations, the MHC mutation rate assumed in the model was very high (~$10^{-1}$). With simulated population sizes of the order of $10^3$–$10^4$, this means that even under neutrality, the expected equilibrium heterozygosity ($H = 4N_e\mu/(1 + 4N_e\mu)$)[8] would be ≈1. In order to investigate more realistic mutation rates, we first reconstructed the model based on the description provided in the methods and supplementary materials of Lighten et al. and verified that we were able to recover their result (Supplementary Fig. 3). We then re-ran the simulations using host mutation rate = $10^{-3}$, but MHC polymorphism was not maintained. When we increased population size to $10^5$ we observed several supertypes but very

few alleles per supertype (Fig. 2a). With a mutation rate = $10^{-4}$, which is in line with the upper limit of reported per locus MHC mutation rates[9,10], all supertype diversity was lost within a few thousand generations. Thus, under more realistic mutation parameters, the model of Lighten et al. does not generate realistic MHC diversity.

Based on their simulations, the authors argue that TSP arises because supertypes are stable in paratope space, and show only slight "wobbles" in the position of their centroids over time. However, our versions of the simulations showed that the long-term stability of supertypes hinges on a crucial feature of the published simulation code that was not mentioned in the verbal model description: in addition to point mutations in pathogens, 1% new parasite genotypes, distributed randomly in epitope space, enter the population at each generation (i.e. ≈100 entirely new, typically very divergent haplotypes in a population of 10,000 individuals). When we used the original parameters of Lighten et al. but 'switched off' host-parasite coevolution by recruiting parasites to the next generation at random (i.e. no selection on parasite genotypes), this seeding alone was sufficient to maintain supertype and allelic diversity similar to that reported by Lighten et al. (Fig. 2b). It thus appears that this easily overlooked feature of unclear biological context (it is debatable whether 100 entirely new, divergent parasite genotypes are likely to enter any population every generation), and not host-parasite coevolution, drove the dynamics of the original simulations.

Without this repeated invasion of so many novel and highly divergent parasite genotypes each generation, we found that host-parasite coevolution led to strikingly different dynamics. All supertype diversity was lost, and there remained a single host supertype chasing a dominant parasite genotype through the whole epitope space (Fig. 2c). However, because Lighten et al. modelled only one parasite species, we investigated whether modelling multiple species could maintain MHC polymorphism

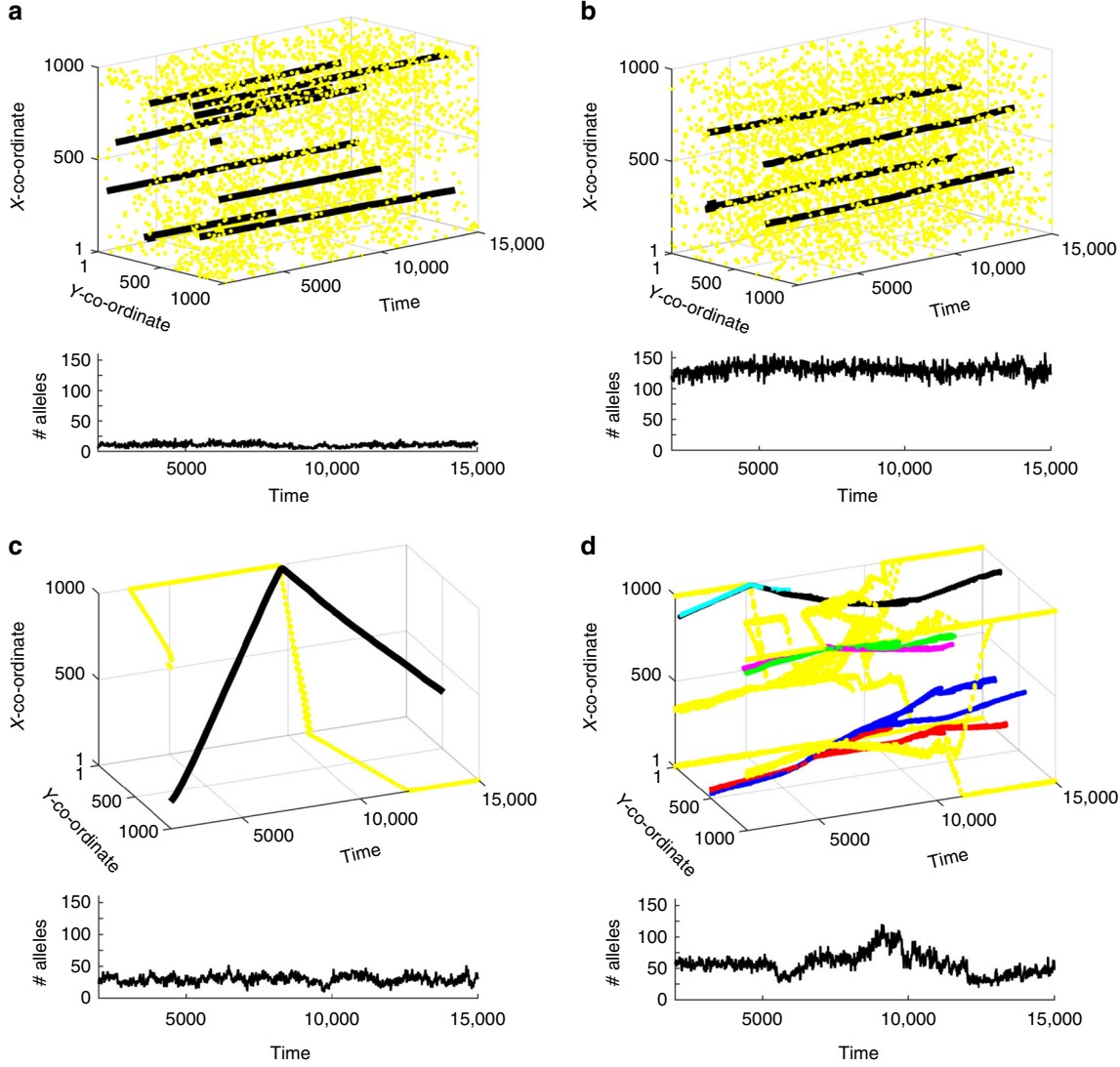

**Fig. 2** Results of simulations based on the model proposed by Lighten et al. Parasite haplotypes (yellow) and host supertypes (black and other colours) are represented as coordinates in 1000 × 1000 grid, reflecting their functional properties (the closer a parasite is to the host, the more likely is a successful host immune response). Each panel shows how changes in specified parameters in the model, compared to the parameters used by Lighten et al. to produce their Fig. 4, affect the outcome (parameters other than stated in the description are as in Lighten et al.). **a** Mutation rate set to $10^{-2}$ for pathogens and $10^{-3}$ for hosts, population size increased to 100,000. In contrast to the results reported by Lighten et al. using much higher mutation rates, the effective number of alleles maintained in a population is small. **b** Selection on parasites (but not on hosts) 'turned off'. This simulation gives the most similar outcome to that reported by Lighten et al., despite the lack of host-parasite coevolution (see Supplementary Fig. 6 for the scenario with no random pathogen genotypes added). **c** Simulations that do not seed each pathogen generation with ≈100 new genotypes, but mutation parameters as in Lighten et al. Host-parasite coevolution utilizing mutational variance alone does not maintain several stable supertypes, even though the simulations started from creating a set of random MHC alleles and pathogens in the same way as Lighten et al. **d** Parameters as in **c**, but 10 independent parasites simulated. Several supertypes are observed at any time point, but they are not stable through time. Effective number of alleles (#alleles) has been calculated for a sample of 100 individuals to allow comparison with the results of Lighten et al. The simulations were written in MatLab and the algorithm follows that described by Lighten et al. The MatLab code is provided in our Supplementary Data 1

and stable persistence of monophyletic supertypes in paratope space without the need for seeding random parasite genotypes. We simulated 10 independent parasite pools, and observed multiple supertypes maintained with realistic numbers of alleles (Fig. 2d). However, the dynamics of these simulations was very different from long-term stability; instead, some supertypes walked large distances in paratope space (implying change in supertype identity), some supertypes were lost, and others branched into daughter supertypes (Fig. 2d). Adding more parasite species did not qualitatively change this conclusion (Supplementary Fig. 4). Supertypes were more stable in paratope space at more realistic mutation rates, but the number of alleles

maintained was very low compared to that observed in natural populations (Supplementary Fig. 5).

The above analyses indicate that under more realistic parameters of host-parasite coevolution, the model of Lighten et al. does not provide support for the scenario of long-term maintenance of stable supertypes, proposed by the authors as an explanation of TSP. We conclude that the proposition of Lighten et al.—that supertypes maintain TSP—is not convincingly supported by empirical evidence, and awaits firm theoretical foundation. What maintains TSP, despite constant pressure for MHC novelty under host-parasite coevolution[3], thus remains an open question.

## Methods

**Phylogenetic analysis**. The relationships between 539 MHC class II exon 2 alleles reported in the Supplementary Data 1 of Lighten et al.[1] were reconstructed from full 209 bp nucleotide sequences using the neighbor Joining method in MEGA 7[11]. The matrix of evolutionary distances was computed using the Jukes–Cantor method. The robustness of the relationships was assessed with 500 bootstrap replicates.

**Population genetics**. We downloaded Lighten et al.'s population-genetic data and analysis scripts from Ben J. Ward's GitHub repository for the paper (https://github.com/BenJWard/Supertypes_RedQueen_TSE) on 17/04/2017, and followed their annotated analysis to produce the data used for Supplementary Fig. 2 and the top-left panel of Supplementary Fig. 1. This analysis first calculates the observed MHC $D_{EST}$[12] between all population pairs in the dataset, and uses each population's mean pairwise $D_{EST}$ as the red, 'observed' dots in the plots. This is followed up by reassigning alleles to supertypes at random, keeping the number of alleles in each supertype constant, and then recalculating each population's mean pairwise $D_{EST}$. The means of 1000 repeats of this randomisation become the blue, 'expected' dots in the respective figures, with the standard deviations used for the blue dots' error bars. Conceptually, the blue dots and their error bars represent a scenario in which there is no selection at the supertype level, i.e. supertype population genetics are a reduced-diversity reflection of allele-based population genetics. An observed supertype-based population structure stronger than this neutral expectation would imply diverging selection on supertypes, whereas lower population structure would imply balancing/stabilising selection.

We then repeated these analyses removing each supertype in turn, i.e. removing all instances of all alleles in the focal supertype, and thereby any individuals that were homozygous for the focal supertype. One population (Cumana, $n = 6$) was homozygous for ST13 and was removed from that supertype's analysis. To compare the effects of removing each supertype, we calculated the mean and SD of the difference between the red and blue dots for each jack-knifed dataset (second column of Supplementary Table 1). We also calculated the Spearman correlation coefficient between each jack-knifed dataset's red dots and the red dots of the full dataset (third column of Supplementary Table 1).

To assess the degree to which each supertype's jack-knifed dataset might be influenced by supertype membership size (number of alleles) or supertype frequency, we performed three additional sets of randomised deletions for each supertype. In the first, we repeatedly removed all occurrences of a random subset of unique nucleotide sequences from the dataset, corresponding in size to the number of sequences in the focal supertype (columns 7–8 of Supplementary Table 1). In the second, we made randomized deletions that matched the frequency of the focal supertype but without systematically removing any particular nucleotide sequences (columns 9–10 of Supplementary Table 1). The third was a spatially structured version of the second, matching the number of random deletions within each population to the frequency of the focal supertype within each population (columns 11–12 of Supplementary Table 1).

**Simulations**. To analyze the theoretical results of Lighten et al. we first reconstructed the simulation model of immune gene evolution presented in their work[1]; the detailed description of the model can be found in the Lighten et al. We then tested their model in an extended parameter space of pathogen mutation rate, host mutation rate and population size, as specified in our figures. Additionally, we ran simulations in three scenarios not considered by Lighten et al. In the first, we removed the feature of Lighten et al.'s simulations that seeded pathogen population with 1% of random genotypes ('random' meaning randomly distributed in physicochemical parameter space) every host generation. The second was a neutral scenario in which parasites were sampled randomly every generation, i.e. without taking into account their ability to infect hosts, thereby preventing parasite adaptation. The third extended simulations of Lighten et al. by including more than one pathogen co-evolving with one host species. The algorithm of the simulation model used in our work can be found in Supplementary Data 1.

**Code availability**. The code for our simulations is available in Supplementary Data 1.

## Data availability

The study used data published by Lighten et al.[1]—see information therein for data availability. Derived datasets and R codes used for the population genetics analyses are available from K.P.P. (karl.p.phillips@gmail.com).

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

## Acknowledgements

We thank Mateusz Konczal, Magda Herdegen-Radwan and Joshka Kaufman for their comments on earlier versions of this manuscript. This research was partly supported by the Jagiellonian University (grants, DS/WB/INoS/756/2018, DS/WB/INoS/757/2018) and Adam Mickiewicz University in Poznan (DS/S/P-B/48).

## Author contributions

J.R. and M.E. designed the simulations and M.E. wrote the code and ran the simulations; W.B. carried out phylogenetic analyses; K.P.P. carried out population-genetic analyses; J.R. drafted the paper with all authors contributing to writing.

## Additional information

**Competing interests:** The authors declare no competing interests.

