## [Peer Review File · Nature Communications]

Reviewers' Comments:

Reviewer #1:

Remarks to the Author:

I have written one set of comments to be shared with both sets of authors, commenting on the Correspondence and Reply separately.

Correspondence: Do MHC supertypes promote trans-species polymorphism?

Maciej J. Ejsmond, Karl P. Phillips, Wieslaw Babik, Jacek Radwan

The correspondence by Ejsmond and colleagues is a critique of the evidence that is used to support the argument of Lighten et al. (2017) that MHC allele supertypes (ST) experience balancing selection while individual alleles experience positive selection. The evidence under debate is the empirical finding of relatively low supertype (ST) population genetic structure and theoretical evidence supplied by simulation modelling used to show that STs promote trans-species polymorphism (TSP).

This correspondence provokes important discussion about balancing selection, the nature of TSP, and the mechanisms generating both. Thus, it is an important contribution to the study of the evolution of the MHC and genes with similar patterns of evolution. I agree with the authors on some points, but not all and I list these in turn below.

L224-29. TSP of supertypes does not necessarily imply a fast turnover of alleles and only rare sharing of alleles between species, and I could not find this argument in Lighten et al. (2017). Instead, they indicate that STs with more functional redundancy may indeed experience a fast turnover, but STs represented by few alleles would be retained and represent that allelic lineage overtime and across species (Lighten et al. pg 8 par 1).

L30-34. Phylogenetic analyses using short fragments of MHC DRB exon 2 (~200bp), with high rates of recombination, gene conversion, and positive selection, can bias phylogenetic interpretation (Schierup & Hein (2000); Posada & Crandall (2002)). Phylogenies based on peptide binding regions (PBR) alone are common among MHC studies, and are often used as an attempt to distinguish between TSP and convergent evolution when compared with nonfunctional regions or alignments of non-PBR sites combined with expected codon usage patterns (e.g. Burri et al. 2010; Lenz et al. 2013; Gillingham et al. 2016). In addition, Lighten et al. (2017) acknowledge that most most supertypes are indeed not monophyletic, and gene conversion (which was absent from their simulation model) could be an explanation (Lighten et al. 2017 Fig. S2).

L36-49. I appreciate the closer examination of ST9 by the authors, and I was myself curious considering it was present in 95% of populations, comprising 55 alleles, and found in 81.64% of individuals. It also has the greatest functional redundancy (Lighten et al. 2017 Fig. S5). The jack-knife removal methods employed by the authors make a compelling case for ST9 driving the relatively low population genetic structuring of STs. I am persuaded that lower genetic structuring of STs is therefore not a general feature of the dataset. However, Lighten et al. (2017) provided 4 lines of evidence supporting that balancing selection was acting on STs (pg 6 par 3):

- 1) Sharing of STs among species (supported)
- 2) Relatively uniform ST distribution across populations (not supported by structure, but partially supported if we accept that similar numbers of STs are shared among populations)
- 3) Lack of correlation with microsatellite differentiation (not supported)
- 4) Deficiency of homozygous STs (supported)

Therefore, there is still evidence that STs experience balancing selection more generally.

L50-66. The reason given by Lighten et al. (2017) for why the MHC mutation rate assumed for the model ($u = \sim 10^{-1}$) was so high was to effectively accelerate evolutionary time in the model (pg 9). Ejsmond et al. provide empirically derived mutation rates from the literature as 10^{-4} , based on studies from inbred mice. I appreciate the authors testing more realistic values of MHC gene mutation rate in their own simulations, and under this lower rate they find that supertype allelic diversity was lost in a few thousand generations, and conclude that the model of Lighten et al. (2017) does not generate realistic MHC diversity. I think this is an important critique.

L67-83. Ejsmond et al. do not allow for overlap in supertype functionality. However, the demarcation of superotypes based on k-means clustering after PCA would depend on the total variation within the population of alleles. Depending on the divergence of groups of alleles in the allele pool, alleles within the same ST with functionally similar properties could indeed have different amino acids at some PBRs.

L84-94. 1% of new parasite genotypes entering the population could be realistic considering the diversity of ubiquitous pathogens a host encounters. This modelling shows that that parasite-mediated selection is sufficient to maintain STs over evolutionary time, independently of host-pathogen coevolution.

L153. unfinished sentence.

References:

Burri, R., Salamin, N., Studer, R.A., Roulin, A. & Fumagalli, L. 2010. Adaptive divergence of ancient gene duplicates in the avian MHC class II β . *Mol. Biol. Evol.* 27: 2360–2374.

Gillingham MAF, Courtiol A, Teixeira M, et al. (2016) Evidence of gene orthology and trans-species polymorphism, but not of parallel evolution, despite high levels of concerted evolution in the major histocompatibility complex of flamingo species. *Journal of Evolutionary Biology* 29, 438-454.

Lenz TL, Eizaguirre C, Kalbe M, Milinski M (2013) EVALUATING PATTERNS OF CONVERGENT EVOLUTION AND TRANS-SPECIES POLYMORPHISM AT MHC IMMUNOGENES IN TWO SYMPATRIC STICKLEBACK SPECIES. *Evolution* 67, 2400-2412.

Posada D, Crandall KA (2002) The Effect of Recombination on the Accuracy of Phylogeny Estimation. *Journal of Molecular Evolution* 54, 396-402.

Schierup MH, Hein J (2000) Consequences of recombination on traditional phylogenetic analysis. *Genetics* 156: 879–891.

Correspondence: Reply to "Do MHC superotypes promote trans-species polymorphism
Cock van Oosterhout & Jackie Lighten

L16-18. The authors write that TSP can be maintained if STs reappear. However, unless alleles related to the original ST lineage evolve to fulfill the lost function, the situation would be one of convergent evolution, not TSP.

L23-35. The authors make the distinction between Red Queen Arms Race (RQAR) and Red Queen Dynamics (RQD), but most work on the evolution of the MHC does not assume two independent

modes of selection. The authors state that the RQAR requires that alleles replace on another in the drive for parasite resistance, but this could also be considered negative frequency dependent selection without ultimate replacement (Slade & McCallum 1992; Apanius et al. 1997).

L41-49. The authors suggest a conceptualization of a battleground of parasite taxa in the perpetual war against parasites (Queen of War, QOW), where supertypes specialize on specific battlegrounds (parasite taxa). While this is a simplification of the process, it does overlook the reality that parasites (and their taxa) comprise many different proteins that can serve as antigenic epitopes that are recognized by MHC molecules. It is extremely unlikely that a single functional supertype will recognize thousands or hundreds of thousands of peptides, and instead much more likely that there will be an overlap in the binding preferences of supertypes. The current conceptualization of the authors implies that supertypes specialize and have non-overlapping functions (i.e., a unit can't fight on multiple battlegrounds at once).

L99-107. The authors build on the simulations of Ejsmond et al for Fig 1 and Fig 2, but it appears that their mutation rate is still unrealistically extremely high ($u=0.9$). I would like to see the same simulations with a more realistic value of u (e.g., between 10^{-3} to 10^{-4}). Would allelic diversity of supertypes still be retained over 1000s of host generations?

References:

Apanius V, Penn D, Slev PR, Ruff LR, Potts WK (1997). The nature of selection on the major histocompatibility complex. *Crit Rev Immunol* 17: 179–224.

Slade RW, McCallum HI (1992). Overdominant vs frequency-dependent selection at MHC loci. *Genetics* 132: 861–862.

Reviewer #2:

Remarks to the Author:

Report on Correspondence: Do MHC supertypes promote trans-species polymorphism? by Ejsmond et al.

This correspondence addresses the article "Evolutionary genetics of immunological supertypes reveals two faces of the Red Queen" by Lighten et al. The authors of the correspondence make two specific points, one disputes the original interpretation of the empirical data regarding the evolution of MHC supertypes in natural populations and one criticizes the simulation model used by Lighten et al. to generalize their findings.

With regard to the first point, the authors here convincingly show that the original interpretation of a lower population differentiation at the supertype level largely hinges on one single supertype (out of 15), and this supertype shows a strikingly different level of genetic variability, suggesting a very special evolutionary history (potentially indicating a non-classical MHC locus). When excluding this particular supertype from the analysis, the population differentiation is not lower than expected by chance, suggesting that supertypes are not maintained in populations for extensive periods of time that would explain the frequently observed trans-species polymorphism at the MHC. However, this is exactly what the original authors had interpreted from their data (neglecting the dominant effect of this one supertype).

With regards to the second criticism, the authors here recapitulate the original simulation and show two important points that severely undermine the original analysis and interpretation. First they point out that the mutation rate at the simulation MHC loci was unrealistically high ($\mu=10^{-1}$). Using instead a mutation rate that is at the upper limit of the realistic range ($\mu=10^{-4}$), the high allelic

diversity and fast turnover (described in the original article as one of the two faces of the Red Queen) breaks down completely. This suggests that the simulation model by Lighten et al. is not suited to simulate realistic scenarios of MHC evolution. Second, the authors point out that the model and its results rely on a continuous seeding of random parasite diversity that is independent of the simulated parasite evolution. This could only be interpreted as some massive migration event from far-away populations in every single generation, which is not very realistic. Even more importantly, the authors of the original article missed to describe this very crucial aspect of the model in the verbal description, which is a serious omission, even if it was unintentional.

I find both points of criticism very relevant and the way they are argued very reasonable. They show that the original article is compromised in its interpretation, and I think that any reader of the original publication should be made aware of this through this correspondence.

I still have a number of comments about the current version of the correspondence manuscript that might help to improve its clarity. They are detailed below.

C1: It would be helpful the general audience to briefly state how supertypes were defined in the original article, and that this same definition was used in the present commentary.

C2: L37: This should probably be "(Fig. 2B, C)", and not "(Fig. 2A, C)". Please check.

C3: Could more evidence be provided that ST9 represents a non-classical locus? For instance, how would a tree look like when combining common alleles of the human classical HLA-DRB1 and the non-classical locus HLA-DMB? Would this look similar to the pattern in Fig. 1B?

C4: It seems to me as if the color coding in Figure 1 is not congruent between panel B and C, at least for ST9. This is confusing. Please consider using the same colors throughout.

C5: In the main text, it says that a population size of 10,000 (10^4) was used to run the models. However in the legend of figure 2 (A) it says that population size was increased to 100,000 (10^5). I suspect that this might affect evolution of allelic diversity. Maybe it's just a typo, but please clarify.

C6: The authors state that when preventing host-parasite coevolution in the simulation, the seeding of new parasites every generation is crucial for maintaining high allelic diversity in the population. However, they only show the plot for the simulation with seeding (Fig. 2B). I think it would be important to also show the plot without the seeding (but still without host-parasite coevolution), to allow for a direct comparison. (Fig 2C is with host-parasite coevolution.)

C7: The authors should explicitly highlight the point that STs are actually lost from the population under certain scenarios without continuous seeding of random parasites. They say that STs are not stable through time, but they should point out the loss more explicitly, since this is the key point that supports their statement, that ST sub-functionalization does not explain TSP per se.

C8: For figure 2C, it is not clear what exactly was done. My understanding is that the simulations by Lighten et al. always started with a randomly distributed set of parasites and alleles. Therefore, shouldn't this at least start in the same way as for instance figure 2A, i.e. with lots of parasite types and alleles/STs? Or were both lost so quickly in the first generations that they don't show up in the plot? This should be described more carefully. I understand that the continuous seeding of novel parasites is odd, but the initial seeding of some random diversity seems reasonable to me. How else should a simulation start without reinventing the Big Bang.

C9: L153: "parameters as in ..."?

C10: I suspect that introducing the possibility of recombination would even further reduce stability of STs over time, especially if recombination happens between alleles of different STs. Maybe this point could be stated too.

C11: I could not see the MatLab code that was promised to be provided in the supplementary material.

C12: The description of the p-values in Table S1 is confusing and I think actually wrong. It seems to me that quantiles of the bootstrap distribution are shown, instead of p-values. Either calculate actual p-values from this (which is simple and preferred) or correct the table labeling.

C13: In Table S1, wouldn't it make more sense to show the average difference between red and blue dots, rather than the correlation of the red dots? I think the lack of correlation in the latter could also be explained by unequal geographic distribution of STs.

RESPONSES TO REVIEWERS' COMMENTS

Reviewer #1 (Remarks to the Author):

I have written one set of comments to be shared with both sets of authors, commenting on the Correspondence and Reply separately.

Correspondence: Do MHC supertypes promote trans-species polymorphism?
Maciej J. Ejsmond, Karl P. Phillips, Wieslaw Babik, Jacek Radwan

The correspondence by Ejsmond and colleagues is a critique of the evidence that is used to support the argument of Lighten et al. (2017) that MHC allele supertypes (ST) experience balancing selection while individual alleles experience positive selection. The evidence under debate is the empirical finding of relatively low supertype (ST) population genetic structure and theoretical evidence supplied by simulation modelling used to show that STs promote trans-species polymorphism (TSP).

This correspondence provokes important discussion about balancing selection, the nature of TSP, and the mechanisms generating both. Thus, it is an important contribution to the study of the evolution of the MHC and genes with similar patterns of evolution. I agree with the authors on some points, but not all and I list these in turn below.

L224-29. TSP of supertypes does not necessarily imply a fast turnover of alleles and only rare sharing of alleles between species, and I could not find this argument in Lighten et al. (2017). Instead, they indicate that STs with more functional redundancy may indeed experience a fast turnover, but STs represented by few alleles would be retained and represent that allelic lineage overtime and across species (Lighten et al. pg 8 par 1).

>> We were not intending to argue that supertype TSP implies fast turnover of alleles – but we see that our original version could have been more clear. What we meant is that (1) TSP is detectable by monophyly of allelic lineages, and (2) that Lighten et al. argue for fast turnover of alleles within supertypes as a general property: “Nevertheless, despite allelic turnover of alleles within ST (Fig 4e), the relative position of each ST remained stable in the epitope/paratope space (Fig.4a, b).”. Given space constraints, we have refined our reasoning to focus on the first part of the argument. The revised version now reads “TSP is a feature of gene genealogy and requires phylogenetic analysis to demonstrate it. In a phylogeny, TSP is detected as monophyletic groups of alleles that occur in descendant species, or by extensive paraphyly should some lineages be lost in either species.”.

L30-34. Phylogenetic analyses using short fragments of MHC DRB exon 2 (~200bp), with high rates of recombination, gene conversion, and positive selection, can bias phylogenetic interpretation (Schierup & Hein (2000); Posada & Crandall (2002)). Phylogenies based on peptide binding regions (PBR) alone are common among MHC studies, and are often used as an attempt to distinguish between TSP and convergent evolution when compared with nonfunctional regions or alignments of non-PBR sites combined with expected codon usage patterns (e.g. Burri et al. 2010; Lenz et al. 2013; Gillingham et al. 2016). In addition, Lighten et al. (2017) acknowledge that most most supertypes are indeed not monophyletic, and gene conversion (which was absent from their simulation model) could be an explanation (Lighten et al. 2017 Fig. S2).

>> We agree that phylogenies based on PBR are useful for some purposes, but whole sequences are better suited to inferring common ancestry – longer sequences are more informative, and interfering processes, like positive selection and micro-recombination, are concentrated in the PBR. We made the following adjustment in the text to address this : “The tree the authors present in their Fig. S2 is based on amino acid sequences at 15 codons under significant positive selection. Although such targeted trees are useful in studying MHC evolution, longer sequences will provide a

more rigorous test of a phylogenetic property such as TSP. We therefore constructed a phylogeny based on Lighten et al.'s full nucleotide sequences (Fig 1B). ”

>>Importantly, TSP is detectable using phylogenetic methods despite the previously described processes that can cause departures from the bifurcating phylogeny. Recombination and other confounding processes thus do not appear strong enough to remove this signal. Therefore, if allelic TSP is due to long-term maintenance of supertypes, supertypes should be predominantly monophyletic or paraphyletic. We hope we have made this point clearer in the revised version (“*Despite the complexities of molecular evolution (recombination, gene conversion) that often cause departures of the true genealogy from a bifurcating tree, such TSP-diagnostic patterns are commonly detected in the MHC (Klein, Garrigan&Hedrick). If TSP is caused by long-term stability of MHC supertypes, the MHC phylogeny of two sister species should be characterized by predominantly monophyletic supertypes (Fig. 1A)*”).

>> Otherwise, if mechanisms, such as recombination distort relationships of alleles within supertypes, making monophyly impossible to detect, such mechanisms should be explicitly included in any theory attempting to explain TSP by long-term stability of supertypes. If these processes make supertypes polyphyletic – regardless of whether they originate via recombination, gene conversion or parallel substitutions (and all these mechanisms are likely) – how can they explain TSP? We have now commented on this in the main text: “*Lighten et al. argue that gene conversion between alleles of different supertypes has broken down monophyly for all supertypes except ST9 (legend to their Fig. S2), but do not explain how polyphyly of supertypes can be reconciled with their proposed role in TSP. Wouldn't different clades of the same supertype fix in different species, even under strong selection maintaining supertypes themselves, erasing TSP?*”.

L36-49. I appreciate the closer examination of ST9 by the authors, and I was myself curious considering it was present in 95% of populations, comprising 55 alleles, and found in 81.64% of individuals. It also has the greatest functional redundancy (Lighten et al. 2017 Fig. S5). The jack-knife removal methods employed by the authors make a compelling case for ST9 driving the relatively low population genetic structuring of STs. I am persuaded that lower genetic structuring of STs is therefore not a general feature of the dataset. However, Lighten et al. (2017) provided 4 lines of evidence supporting that balancing selection was acting on STs (pg 6 par 3):

>> When set against the influence of ST9, we feel these lines of evidence are not very convincing (see below) and, importantly, that they do not imply that this is supertypes, and not alleles, that are under balancing selection.

1) Sharing of STs among species (supported)

>> Yes, this is a suggestive argument, although the authors have not demonstrated that this sharing is higher than expected under ‘traditional’ balancing selection acting on MHC alleles. Given that STs are nowhere near monophyletic, we don’t see how their model is an improvement on ‘traditional’, allelic balancing selection as an explainer of TSP.

2) Relatively uniform ST distribution across populations (not supported by structure, but partially supported if we accept that similar numbers of STs are shared among populations)

>> We agree that the explicit test (more uniform ST distribution than that of alleles) does not support this. Yes, the similar numbers of STs among populations may be indicative of such a process, but Lighten et al. do not test whether this pattern can be explained by STs being a less diverse marker than alleles.

3) Lack of correlation with microsatellite differentiation (not supported)

>> Agreed, no support

4) Deficiency of homozygous STs (supported)

>> However, even though the authors cannot assign alleles to loci, they “assumed for this analysis that STs are locus specific”. We agree that some supertypes may be locus-specific to varying degrees, but the authors do not provide strong evidence against supertypes being shared among loci. They report that 1600 out of 1675 fish carried no more than two alleles (which is 95%, not 99% as they report), but this implies that at least some supertypes are shared between loci (probably well over 5%, as an unknown number of cases would remain undetected). This very much weakens the “deficiency of ST homozygotes” argument. Additionally, the ‘ST homozygote’ argument is heavily influenced by ST9 – 6.7% of individuals are ST homozygotes in the full dataset, but 29.1% are homozygotes if ST9 and all its alleles are removed. We have now commented on this in the manuscript: *“Another argument which Lighten et al. used to support balancing selection acting on supertypes, i.e. deficit of “supertype homozygotes”, is unconvincing because the authors could not assign alleles to loci (genotypes consisted of up to nine alleles, implying a minimum of five co-amplifying loci in some individuals), so that true zygosity is not known. Furthermore, the unique properties of ST9 also mean that the observed deficit needs to be treated cautiously (6.7% of individuals are ST homozygotes in the full dataset, but 29.1% if ST9 and all its alleles are removed).”*

Therefore, there is still evidence that STs experience balancing selection more generally.

>> Given the arguments above, we do not feel that Lighten et al. make a good case that balancing selection acting on supertypes is stronger than balancing selection acting on MHC alleles.

L50-66. The reason given by Lighten et al. (2017) for why the MHC mutation rate assumed for the model ($\mu = \sim 10^{-1}$) was so high was to effectively accelerate evolutionary time in the model (pg 9). Ejsmond et al. provide empirically derived mutation rates from the literature as 10^{-4} , based on studies from inbred mice. I appreciate the authors testing more realistic values of MHC gene mutation rate in their own simulations, and under this lower rate they find that supertype allelic diversity was lost in a few thousand generations, and conclude that the model of Lighten et al. (2017) does not generate realistic MHC diversity. I think this is an important critique.

>> Thank you for appreciating this.

L67-83. Ejsmond et al. do not allow for overlap in supertype functionality. However, the demarcation of supertypes based on k-means clustering after PCA would depend on the total variation within the population of alleles. Depending on the divergence of groups of alleles in the allele pool, alleles within the same ST with functionally similar properties could indeed have different amino acids at some PBRs.

>> We intended to point out that the proposed stability of supertypes implies functional constraints on antigen-binding sites, whose physicochemical descriptors define supertype identity. Such a constraint must have an effect on non-synonymous substitution rates at ABSs, although the magnitude of this effect is unclear. A proper assessment of this would require a self-standing study, so we have decided to withdraw this argument from our revised manuscript.

L84-94. 1% of new parasite genotypes entering the population could be realistic considering the diversity of ubiquitous pathogens a host encounters. This modelling shows that that parasite-mediated selection is sufficient to maintain STs over evolutionary time, independently of host-pathogen coevolution.

>> Yes, we agree that this ‘1% of new parasite genotypes’ figure could be realistic in itself, but we do not think that the properties of these genotypes as modelled by Lighten et al. are. In their model, each immigrant is a distinct, randomly created genotype. In reality, immigrant parasites from neighbouring populations would carry a sample of the limited range of genotypes prevalent in those populations, and would not be entirely new each generation. So we think scenario modelled by Lighten et al. is rather extreme, if not unrealistic – a view shared by Reviewer 2. Please also note that in their response, Lighten et al. consider seeding as “emerging infectious diseases”, which, at

this rate, would almost certainly be unrealistic. Nevertheless, we decided that the adjective “unrealistic” is not necessary to make our point, and have removed it to avoid controversy.

L153. unfinished sentence.

>> Our apologies: should read “as in Lighten et al.”. Now corrected

Reviewer #2 (Remarks to the Author):

Report on Correspondence: Do MHC supertypes promote trans-species polymorphism? by Ejsmond et al.

This correspondence addresses the article “Evolutionary genetics of immunological supertypes reveals two faces of the Red Queen” by Lighten et al. The authors of the correspondence make two specific points, one disputes the original interpretation of the empirical data regarding the evolution of MHC supertypes in natural populations and one criticizes the simulation model used by Lighten et al. to generalize their findings.

With regard to the first point, the authors here convincingly show that the original interpretation of a lower population differentiation at the supertype level largely hinges on one single supertype (out of 15), and this supertype shows a strikingly different level of genetic variability, suggesting a very special evolutionary history (potentially indicating a non-classical MHC locus). When excluding this particular supertype from the analysis, the population differentiation is not lower than expected by chance, suggesting that supertypes are not maintained in populations for extensive periods of time that would explain the frequently observed trans-species polymorphism at the MHC. However, this is exactly what the original authors had interpreted from their data (neglecting the dominant effect of this one supertype).

With regards to the second criticism, the authors here recapitulate the original simulation and show two important points that severely undermine the original analysis and interpretation. First they point out that the mutation rate at the simulation MHC loci was unrealistically high ($\mu=10^{-1}$). Using instead a mutation rate that is at the upper limit of the realistic range ($\mu=10^{-4}$), the high allelic diversity and fast turnover (described in the original article as one of the two faces of the Red Queen) breaks down completely. This suggests that the simulation model by Lighten et al. is not suited to simulate realistic scenarios of MHC evolution. Second, the authors point out that the model and its results rely on a continuous seeding of random parasite diversity that is independent of the simulated parasite evolution. This could only be interpreted as some massive migration event from far-away populations in every single generation, which is not very realistic. Even more importantly, the authors of the original article missed to describe this very crucial aspect of the model in the verbal description, which is a serious omission, even if it was unintentional. I find both points of criticism very relevant and the way they are argued very reasonable. They show that the original article is compromised in its interpretation, and I think that any reader of the original publication should be made aware of this through this correspondence. I still have a number of comments about the current version of the correspondence manuscript that might help to improve its clarity. They are detailed below.

>> Thank you for appreciating our effort.

C1: It would be helpful the general audience to briefly state how supertypes were defined in the original article, and that this same definition was used in the present commentary.

>> We have now defined supertypes in the first paragraph by adding “...‘supertypes’ (clusters of MHC alleles with similar physicochemical properties at their antigen-binding sites)”. This is the same definition used by Lighten et al.

C2: L37: This should probably be “(Fig. 2B, C)”, and not “(Fig. 2A, C)”. Please check.

>> Yes, thanks for pointing this out

C3: Could more evidence be provided that ST9 represents a non-classical locus? For instance, how would a tree look like when combining common alleles of the human classical HLA-DRB1 and the non-classical locus HLA-DMB? Would this look similar to the pattern in Fig. 1B?

>> Following the Reviewer’s suggestion, we looked at human DMB and DOB genes. However, these genes show extremely low polymorphism, with only a handful of common alleles, most of which have identical exon 2 sequences. The pattern of nonclassical HLA class II polymorphism is thus very different from that of ST9. Please note, however, that we do not insist that ST9 represents a nonclassical locus or a pseudogene – we just float them as possibilities. The take-home message in our ST9 argument is that patterns of genetic differentiation and the shape of ST9 genealogy both depart drastically from what is observed in the other STs. The factors underlying these differences are unclear but strongly suggest that mechanisms affecting ST9’s evolution are different from mechanisms driving evolution of the other STs. To avoid confusion, in the revised version we have removed speculations about possible pseudogene or nonclassical status of ST9.

C4: It seems to me as if the color coding in Figure 1 is not congruent between panel B and C, at least for ST9. This is confusing. Please consider using the same colors throughout.

>> We’ve decided to withdraw the original part B of the figure, as we now do not discuss predictions from fast allele turnover as implied by Lighen et al. (see response to first point of Reviewer 1).

C5: In the main text, it says that a population size of 10,000 (10^4) was used to run the models. However in the legend of figure 2 (A) it says that population size was increased to 100,000 (10^5). I suspect that this might affect evolution of allelic diversity. Maybe it’s just a typo, but please clarify.

>> Yes, in these simulations we used larger populations. We did this because at more realistic mutation rates, populations of 10,000 did not maintain multiple supertypes. While we stated the increase in the figure description (“population size increased to 100 000”), we failed to provide justification in the main text – a silly oversight, as the need to raise the population size strengthens our argument. We have now amended this: “We then re-ran the simulations using host mutation rate = 10^{-3} , but MHC polymorphism was not maintained. When we increased population size to 10^5 , we observed several supertypes but very few alleles per supertype (Fig. 2A),...”

C6: The authors state that when preventing host-parasite coevolution in the simulation, the seeding of new parasites every generation is crucial for maintaining high allelic diversity in the population. However, they only show the plot for the simulation with seeding (Fig. 2B). I think it would be important to also show the plot without the seeding (but still without host-parasite coevolution), to allow for a direct comparison. (Fig 2C is with host-parasite coevolution.)

>> We have now added simulations of the scenario requested by the reviewer (Supplementary materials, Fig. S6; we cannot expand Fig. 2 without losing readability, and we are limited to two figures in the main text). We have also added two more supplementary figures (S4 and S5) to make parameter space we explored more complete, and added the following comment in the main text *“Adding more parasite species did not qualitatively change this conclusion (Supplementary materials, Fig. S4). Supertypes were more stable in paratope space at more realistic mutation rates, but the number of alleles maintained was very low compared to that observed in natural populations (Supplementary materials, Fig. S5).”*

C7: The authors should explicitly highlight the point that STs are actually lost from the population under certain scenarios without continuous seeding of random parasites. They say that STs are not stable through time, but they should point out the loss more explicitly, since this is the key point that supports their statement, that ST sub-functionalization does not explain TSP per se.

>> Good point. We have now stressed that supertype diversity is lost without seeding *“Without this repeated invasion of so many novel and highly divergent parasite genotypes each generation, we found that host-parasite coevolution led to strikingly different dynamics. All supertype diversity was lost, and there remained a single host supertype chasing a dominant parasite genotype through the whole epitope space (Fig. 2C)”*

C8: For figure 2C, it is not clear what exactly was done. My understanding is that the simulations by Lighten et al. always started with a randomly distributed set of parasites and alleles. Therefore, shouldn't this at least start in the same way as for instance figure 2A, i.e. with lots of parasite types and alleles/STs? Or were both lost so quickly in the first generations that they don't show up in the plot? This should be described more carefully. I understand that the continuous seeding of novel parasites is odd, but the initial seeding of some random diversity seems reasonable to me. How else should a simulation start without reinventing the Big Bang.

>> Yes, we started all simulations in the same way as Lighten et al. did. While we tried to convey this in the main text and figure 2 legend, we now state it more explicitly in Fig 2C's legend: *“Host-parasite coevolution utilizing mutational variance alone does not maintain several stable supertypes, even though the simulations started from creating a set of random MHC alleles and pathogens in the same way as Lighten et al.”* – we hope it is now clear that more supertypes were present at the outset of simulations but that they quickly disappeared.

C9: L153: “parameters as in ...”?

>> “...as in Lighten et al.”; apologies for overlooking this. Now amended

C10: I suspect that introducing the possibility of recombination would even further reduce stability of STs over time, especially if recombination happens between alleles of different STs. Maybe this point could be stated too.

>> We agree, but we are reluctant to state this without proper supporting analysis; recombination cannot be easily implemented in the framework of Lighten et al. (there are no MHC sequences, just two coordinates for each allele), and such additional analysis would be beyond the scope of the present Correspondence

C11: I could not see the MatLab code that was promised to be provided in the supplementary material.

>> Our apologies. We did send the code to Jackie Lighten during our pre-submission correspondence, and we meant to include it during submission. We'll make sure to include it with the revision!

C12: The description of the p-values in Table S1 is confusing and I think actually wrong. It seems to me that quantiles of the bootstrap distribution are shown, instead of p-values. Either calculate actual p-values from this (which is simple and preferred) or correct the table labeling.

>> Yes, the reviewer's reading of the table is correct, and we see it could have been better presented. We have changed the column headers from 'P' to 'Quantile', and updated the table's legend.

C13: In Table S1, wouldn't it make more sense to show the average difference between red and blue dots, rather than the correlation of the red dots? I think the lack of correlation in the latter could also be explained by unequal geographic distribution of STs.

>> We have added an extra column to the table ('Mean obs.-exp. diff. (\pm SD)') with the analysis that the reviewer suggested, and we have made this the first column to maximize its prominence. However, we have also retained the red-v-red correlations because they still help explore the effects we present, and generating blue dots with every resample would be too computationally intensive. Yes, the much weaker red-v-red correlation for ST9 can be explained by the unequal geographic distribution of STs, but this is part of the point we're trying to make – ST9 exerts a disproportionate effect on population genetic structure assessed by STs, and that one of the paper's key empirical observations (an important observation in ecological MHC research!) is thus not a general property of the dataset.

>> POSTSCRIPT: COMMENTS ON SOME ISSUES RAISED BY LIGHEN ET AL. IN THE REPLY.

- 1) Lighten et al. claim that we misunderstood their hypothesis and modelled a single parasite chasing a single host in epitope space. This is not the case: Fig 2C is an outcome and not a starting point. Its starting point is exactly the same as that for Fig 2A, with the only difference being the switching off of repeated seeding with random parasite genotypes. We tried to make it clearer in the revised version.
- 2) Lighten et al. respond: "Contra Ejsmond et al., our model does not imply the monophyly of STs. Rather, the QOW will result in long terminal branches in the phylogeny of the MHC, and the sharing of ancient branches between species (i.e. TSP). Polyphyletic STs can occur because of recombination between distinct branches, and conversely, a single ST lineage can bifurcate to increase the coverage of the epitope space". We found this statement to be unclear. Firstly, the authors don't specify a mechanism that can generate long terminal branches (long terminal branches are not explicitly mentioned in the original paper). If long terminal branches were the result of rapid turnover of alleles within supertypes, then we would expect limited variation within supertypes and little allele sharing even between closely related species, which is not the case (Fig. 1B). Secondly, what are the ancient branches shared between species if not supertypes? If they are not supertypes, then how are supertypes maintaining TSP? Finally, how can polyphyletic supertypes - whether they originate via recombination, gene conversion or parallel substitutions (and all these mechanisms are likely) – explain TSP? What does a 'bifurcating ST' mean in phylogenetic

terms, and how does it square with the authors' assertion that supertypes are relatively stable? Does this imply a hierarchy of supertypes? Again, this has not been mentioned in the original paper.

- 3) **Lighten et al. respond:** "Furthermore, and contra Ejsmond et al., the sharing of identical alleles does not falsify the rapid allelic turnover suggested by our model. Species can share identical alleles through balancing selection, incomplete lineage sorting...". **We cannot understand how fast allele turnover can be consistent with incomplete lineage sorting. Furthermore, it seems to be the very essence of the Lighten et al. paper that balancing selection acts on supertypes, and not on alleles. Although we don't think that they consider their model to be the exclusive driver of MHC evolution, this type of balancing selection is supposed not to be part of the authors' Red Queen Arms Race scenario.**

- 4) **Lighten et al. respond:** 7) "Ejsmond et al.'s suggestion that the population structuring of other STs falsifies our model is incorrect as well; the guppy is infected by ~70 species, and populations across Trinidad are complex and highly diverged in parasite fauna (Stanley King, Pers. Comm.). The QOW is fighting battles at different fronts in the different guppy populations, which in turn inflates the differentiation of STs at the population level. Across the gene pools of species, however, we observe the same STs, with *M. picta* sharing all STs with *P. reticulata*." **This implies that Lighten et al.'s hypothesis is not falsifiable – that it can produce both decreased and increased population genetic structure. They also seem to have changed their minds on how they're reading population genetic structure by supertypes, as in the original article they say:** "In summary, population genetic analysis shows that ST variation is subject to balancing selection, as is evidenced by... the relatively uniform ST distribution across populations"; **and,** "In our study, we observed a relative uniform ST distribution across populations, as well as a deficiency of 'homozygous STs'; population genetic signatures consistent with balancing selection"; **and,** "In other words, the simulations show that the observed ST distribution across populations is too uniform to be explained by a random process such as genetic drift, but suggests that balancing selection is homogenising the ST diversity across the guppy populations"

- 5) **The Reply concludes:** "Conversely to Ejsmond et al., we 'switched off' the RQAR after 5000 generations (i.e. parasites at generation t did not reproduce offspring in $t+1$), and we only introduced random novel parasites every generation. Figure 2c illustrates that TSP can arise also without the RQAR, just by the input of novel parasites. This shows that the Queen of War, not a single Red Queen Arms Race, maintains TSP. The Queen is dead, long live the Queen". **We had to do a double-take on this statement! Firstly, they seem to have overlooked that this is one of the issues we raise: "1% seeding alone was sufficient to maintain supertype and allelic diversity similar to that reported by Lighten et al. (Fig. 2B). It thus appears that this easily overlooked feature of unclear biological rationale, and not host-parasite coevolution, drove the dynamics of the original simulations" (Ejsmond et al. manuscript, lines 81-83; wording has changed slightly in the revised version). Secondly, as we argued in our Correspondence, this means that the stability of supertypes in their simulations is NOT driven by coevolution. Are the authors now saying that this is what they meant all along? If so, this is not clear in the original paper: "Evolutionary genetics of**

immunological supertypes reveals two faces of the Red Queen” (**original paper’s title**); and, “Red Queen host–parasite co-evolution can drive adaptations of immune genes by positive selection that ... results in a balanced polymorphism (Red Queen dynamics) and long-term preservation of genetic variation (trans-species polymorphism)” (**abstract of paper**); and, “We observed significant fluctuations in ST frequency in response to changes in parasite frequency over time (Fig. 4c), consistent with balancing selection (Red Queen dynamics)” (**main body of paper, page 6, column 1**); and, in their Reply, “Red Queen Dynamics (RQD) operates at the coevolutionary interaction between two antagonistic species, resulting in a balanced polymorphism” (**first paragraph**).

- 6) Lighten et al. extend our simulations with more than one pathogen species and report that supertypes remain stable with > 25 parasites. When we extended our simulations to higher number of parasites (Supplementary Fig. 5), stability occurred only in occasional runs.

Reviewers' Comments:

Reviewer #1:

Remarks to the Author:

Correspondence: Do MHC supertypes promote trans-species polymorphism?

Maciej J. Ejsmond, Karl P. Phillips, Wieslaw Babik, Jacek Radwan

I am very satisfied with the responses of the authors to the points raised by the reviewers and the Reply. I believe this Correspondence is an important contribution to the debate on MHC evolution. I have a couple minor points that I mention for thoroughness. Line numbers come from my adding them to the documents in word.

L61-63. "If TSP is caused by long-term stability of MHC supertypes, the MHC phylogeny of two sister species should be characterized by predominantly monophyletic supertypes (Fig. 1A)".

I agree with the authors in this paragraph (L54-63), but also want to highlight that TSP is not observed for all allelic lineages between groups of species. It does commonly occur, but not always. There are many cases of alleles grouping by species, and not across species. I have the impression that the authors are arguing that if all STs are not monophyletic, then this cannot be an explanation for TSP more generally.

L390-393. "Lighten et al. extend our simulations with more than one pathogen species and report that supertypes remain stable with > 25 parasites. When we extended our simulations to higher number of parasites (Supplementary Fig. 5), stability occurred only in occasional runs."

Interesting point. Can this be highlighted in the correspondence?

Fig. 2 Caption. Should Lighten et al. 2014 be Lighten et al. 2017?

Reviewer #2:

Remarks to the Author:

Report on revised correspondence by Ejsmond et al.

I have read the response to my comments and the revised correspondence. I am very happy to see that the authors have taken great care to address my points as well as the points by the other reviewer. I think this effort has improved the correspondence even further, which now makes a very clear and convincing argument against two central claims in the article by Lighten et al.

I have no further comments or concerns about the correspondence by Ejsmond et al. and would be delighted to see it published as soon as possible, in order to limit further dissemination of the problematic claims from Lighten et al.

REVIEWERS' COMMENTS:

Reviewer #1 (Remarks to the Author):

Correspondence: Do MHC supertypes promote trans-species polymorphism?
Maciej J. Ejsmond, Karl P. Phillips, Wieslaw Babik, Jacek Radwan

I am very satisfied with the responses of the authors to the points raised by the reviewers and the Reply. I believe this Correspondence is an important contribution to the debate on MHC evolution. I have a couple minor points that I mention for thoroughness. Line numbers come from my adding them to the documents in word.

L61-63. "If TSP is caused by long-term stability of MHC supertypes, the MHC phylogeny of two sister species should be characterized by predominantly monophyletic supertypes (Fig. 1A)".

I agree with the authors in this paragraph (L54-63), but also want to highlight that TSP is not observed for all allelic lineages between groups of species. It does commonly occur, but not always. There are many cases of alleles grouping by species, and not across species. I have the impression that the authors are arguing that if all STs are not monophyletic, then this cannot be an explanation for TSP more generally.

>> Indeed, we think that most supertypes should be monophyletic, if their presence is to explain TSP. Otherwise, if the same supertype originated convergently multiple times, the pattern of TSP would not be expected. However, a monophyletic supertype may be lost in one or more species – this will not affect monophyly, but will lead to some allelic lineages being present only in one species. Assuming long-term maintenance of supertypes, such situations should be rarer than loss of individual alleles. The paragraph has been reworded to clarify (lines 29-30 and 34-45).

L390-393. "Lighten et al. extend our simulations with more than one pathogen species and report that supertypes remain stable with > 25 parasites. When we extended our simulations to higher number of parasites (Supplementary Fig. 5), stability occurred only in occasional runs."

Interesting point. Can this be highlighted in the correspondence?

>> This referred to simulations in Lighten's et al. Reply to our comments. The Editor notified us that the response will not be published.

Fig. 2 Caption. Should Lighten et al. 2014 be Lighten et al. 2017?

>> Yes, our apologies, this should be Lighten et al. 2017 – now corrected.

Reviewer #2 (Remarks to the Author):

Report on revised correspondence by Ejsmond et al.

I have read the response to my comments and the revised correspondence. I am very happy to see that the authors have taken great care to address my points as well as the points by the other reviewer. I think this effort has improved the correspondence even further, which now makes a very clear and convincing argument against two central claims in the article by Lighten et al.

I have no further comments or concerns about the correspondence by Ejsmond et al. and would be delighted to see it published as soon as possible, in order to limit further dissemination of the problematic claims from Lighten et al.

>> We would like to thank both reviewers for their comments and for appreciating our effort!